# *This* Reads Like *That*: Deep Learning for Interpretable Natural Language Processing

**Claudio Fanconi**[*]
ETH Zürich
fanconic@ethz.ch

**Moritz Vandenhirtz**[*]
ETH Zürich
moritz.vandenhirtz@inf.ethz.ch

**Severin Husmann**
ETH Zürich
shusmann@ethz.ch

**Julia E. Vogt**
ETH Zürich
julia.vogt@inf.ethz.ch

## Abstract

Prototype learning, a popular machine learning method designed for inherently interpretable decisions, leverages similarities to learned prototypes for classifying new data. While it is mainly applied in computer vision, in this work, we build upon prior research and further explore the extension of prototypical networks to natural language processing. We introduce a learned weighted similarity measure that enhances the similarity computation by focusing on informative dimensions of pre-trained sentence embeddings. Additionally, we propose a post-hoc explainability mechanism that extracts prediction-relevant words from both the prototype and input sentences. Finally, we empirically demonstrate that our proposed method not only improves predictive performance on the AG News and RT Polarity datasets over a previous prototype-based approach, but also improves the faithfulness of explanations compared to rationale-based recurrent convolutions.

## 1 Introduction

Due to the increasing utilization of artificial neural networks in critical domains such as medicine and autonomous driving, the need for interpretable and explainable decisions made by these models is extremely important (Samek and Müller, 2019; Rudin, 2019). While extensive research in this field has focused on post-hoc explanations for black-box machine learning models (Bach et al., 2015; Kim et al., 2018; Bau et al., 2017; Lapuschkin et al., 2019), a recent line of investigation has challenged this paradigm by proposing network architectures that inherently possess interpretability (Girdhar and Ramanan, 2017; Chen et al., 2019; Li et al., 2018; Brendel and Bethge, 2019). These interpretable models aim to base their decisions on a limited set of human-understandable features, enabling users to comprehend the decision-making process while maintaining high predictive performance (Chen et al., 2019).

In this work, we focus on transforming and enhancing the prototype-based model introduced by Chen et al. (2019) from the computer vision domain to the natural language processing (NLP) domain. Prototype-based methods for NLP, as explored in previous works (Friedrich et al., 2022; Ming et al., 2019; Huang et al., 2022; Wang et al., 2022), involve classifying test samples based on their similarity to prototypical training samples. To improve upon these prior adaptations, we propose two key enhancements. Firstly, we introduce a learned weighted similarity measure that enables the model to attend to the most informative aspects of the pre-trained sentence embedding. This improvement ensures that the method can benefit from pre-trained language models (Reimers and Gurevych, 2019), which have shown superb empirical results, and extract task-relevant information, thereby enhancing its performance. Secondly, we enhance the interpretability mechanism by conducting post-hoc analyses to determine the importance of individual words in both test samples and sentence prototypes. Furthermore, we investigate the faithfulness of these explanations, examining their ability to faithfully capture and represent the underlying model's decision-making process.

In summary, our contributions in this paper involve domain-specific adaptations of inherently interpretable prototype-based models from computer vision to the NLP domain, enhancing their performance through a learned weighted similarity measure, improving explainability via post-hoc analyses of word importance, and investigating the faithfulness thereof. These advancements aim to further the understanding and trustworthiness of prototype-based methods in NLP. [1]

---

[*]Equal contribution

[1]The code is publicly available at https://github.com/fanconic/this_reads_like_that.

## 2 Related Work

The three main interpretability mechanisms for deep neural networks discussed in this work are post-hoc explainability, rationale-based models, and prototype-based models.

Post-hoc explanation methods correlate inputs and outputs to explain a model's behavior using gradient visualization (Sundararajan et al., 2017; Smilkov et al., 2017) and attention mechanisms (Bahdanau et al., 2015; Li et al., 2017; Husmann et al., 2022; Abnar and Zuidema, 2020). However, the faithfulness of these methods is uncertain, as learned attention weights can lack correlation with gradient-based measures of feature importance, and different attention distributions can yield equivalent predictions (Jain and Wallace, 2019).

Rationale-based models (Lei et al., 2016; Jain et al., 2020; Bastings et al., 2019) improve explanation faithfulness by selecting rationales that serve as explanations for predictions. While rationales indicate contributing input features, they do not reveal how these features are combined or processed within the neural network.

Prototype-based networks (Chen et al., 2019) emulate human-like reasoning by finding similarities between input parts and prototypes to make predictions. However, existing NLP-based prototype methods (Friedrich et al., 2022; Ming et al., 2019; Huang et al., 2022; Wang et al., 2022) require full model control and lack the ability to visualize important words in sentences, failing the criterion of sparse explanations (Rudin, 2019). Our proposed method addresses these limitations. Further details on related work can be found in Appendix A.1.

## 3 Methods

In our approach, we first compute the sentence embedding using a pretrained language models BERT (Reimers and Gurevych, 2019), GPT2 (Radford et al., 2019; Muennighoff, 2022), MPNet (Song et al., 2020), and RoBERTa (Sanh et al., 2019) to capture the semantic representation of the input sentence. Then, we utilize a prototypical layer to measure the similarity between the sentence embedding and learned prototypes. These prototypes act as learned prototypical points in the embedding space, representing the different classes, to which they have been assigned equally. The similarity scores are multiplied by the learned prototype weights to arrive at the final, inherently interpretable prediction.

To provide a meaningful interpretation in the input space, we periodically project the prototypes to their nearest training sample of the same class. For the last three epochs, the prototypes are projected and fixed, thus enabling visualization of the prototypes contributing to an input's prediction at test time. To ensure the useful spreading of prototypes, we employ the loss terms proposed by Friedrich et al. (2022). In Appendix A.4, we conduct an ablation study to assess the effect of the different terms. More information about the experimental setup and implementation can be found in Appendix A.2.

### 3.1 Learned Weighted Similarity Measure

Wanting to leverage the strong empirical performance of pre-trained language models, the issue arises that their sentence embeddings contain information relevant to multiple tasks (Reimers and Gurevych, 2019), whereas we are only interested in one (e.g. sentiment classification). Thus, the similarity to prototypes, that is computed directly on these pre-trained embeddings, should only depend on class-relevant information. To achieve this, we propose a new similarity measure that assigns individual weights to the dimensions of the embedding, inspired by the work of Vargas and Cotterell (2020) on gender bias as a linear subspace in embeddings. This allows the network to focus on dimensions it deems more important for our specific task, rather than treating all dimensions equally. In this measure, we calculate the similarity (i.e. $\ell_2$ distance or cosine similarity) between the embeddings and prototypes, considering the weights assigned to each dimension. For cosine similarity, we multiply each dimension of the embeddings by the corresponding learned weight, as shown in Equation 1:

$$\text{sim}(\mathbf{w}, \mathbf{u}, \mathbf{v}) = \frac{\sum_i w_i u_i v_i}{\sqrt{\sum_i w_i u_i^2}\sqrt{\sum_i w_i v_i^2}}, \quad (1)$$

where $\mathbf{w}$ represents the weight vector, and $\mathbf{u}$ and $\mathbf{v}$ are the embeddings whose similarity is being calculated. During training, we enforce non-negative reasoning by clamping the weights to a minimum of 0. This ensures that negative reasoning is disallowed. Similarly, for the $\ell_2$ similarity, we weigh the embedding dimensions according to the weight vector:

$$\text{sim}(\mathbf{w}, \mathbf{u}, \mathbf{v}) = \sqrt{\sum_i (w_i u_i v_i)^2} \quad (2)$$

## 3.2 Visualizing Important Words

Instead of the sentence-level approach proposed by Friedrich et al. (2022), which visualizes the whole prototypes as explanation, we introduce a word-level visualization for improved interpretability through sparse explanations (Rudin, 2019). On a high level, we visualize the words of a sample that were most important for the similarity to the set of prototypes, which had the biggest effect on the prediction. This is achieved by computing the discrete derivative of the similarity to these prototypes with respect to each input token. We iteratively remove the token from the input that decreases the similarity the most, as upon reversion, this token is most important for the similarity. Likewise, for visualizing important words of prototypes, we consider the similarity to the input sample. By iteratively selecting and removing the tokens, we visualize the most important words for a given prototype and test sample. The former can be interpreted as a form of abstractive rationales (novel words are generated to support the prediction), while the latter are extractive rationales (important words from the input data are extracted to support the prediction) (Gurrapu et al., 2023).

Unlike previous approaches (Friedrich et al., 2022), we do not impose restrictions on the relative position of the words or fix the number of visualized words in advance. Instead, we dynamically determine the number of words to be visualized based on the explained percentage of the full distance. We define full distance as the change in similarity after having removed the top $n$ tokens (where $n = 10$), hereby implicitly assuming that the similarity between any two sentences can be erased by removing $n$ tokens. Then, we visualize the smallest subset of removed tokens that together account for at least $q \in [0, 1]$ (in our case $q = 0.75$) of the full distance. This approach provides more flexibility and adaptability in visualizing the important words of a sentence and its prototypes without requiring predefined specifications of their structure. It offers a more nuanced understanding of the reasons behind the similarity between a test sample and the prototypes and thus of the decision-making process for predicting a sample's label.

## 4 Results

In this section, we present experiments demonstrating the importance of our proposed method. In the first experiment, the models were run five times

| Backbone | Similarity | Dataset | |
| | | RT reviews | AG News |
|---|---|---|---|
| BERT | cosine | $80.46 \pm 0.43$ | $78.67 \pm 0.21$ |
| | w. cosine | $\mathbf{82.14} \pm 0.20$ | $\mathbf{86.13} \pm 0.12$ |
| | $\ell_2$ distance | $79.20 \pm 0.36$ | $\mathbf{73.36} \pm 0.89$ |
| | w. $\ell_2$ distance | $\mathbf{80.02} \pm 0.44$ | $73.11 \pm 0.82$ |
| GPT2 | cosine | $73.41 \pm 0.41$ | $85.77 \pm 0.20$ |
| | w. cosine | $\mathbf{75.44} \pm 0.05$ | $\mathbf{87.25} \pm 0.13$ |
| | $\ell_2$ distance | $69.39 \pm 1.78$ | $78.36 \pm 1.46$ |
| | w. $\ell_2$ distance | $\mathbf{69.88} \pm 0.46$ | $\mathbf{79.36} \pm 1.36$ |
| MPNet | cosine | $\mathbf{81.09} \pm 0.49$ | $86.49 \pm 0.32$ |
| | w. cosine | $79.55 \pm 0.82$ | $\mathbf{88.09} \pm 0.28$ |
| | $\ell_2$ distance | $75.87 \pm 4.21$ | $\mathbf{79.84} \pm 1.50$ |
| | w. $\ell_2$ distance | $\mathbf{77.69} \pm 2.26$ | $79.73 \pm 1.38$ |
| RoBERTa | cosine | $75.58 \pm 2.37$ | $86.24 \pm 0.06$ |
| | w. cosine | $\mathbf{77.51} \pm 0.29$ | $\mathbf{87.45} \pm 0.07$ |
| | $\ell_2$ distance | $72.57 \pm 1.79$ | $80.48 \pm 0.90$ |
| | w. $\ell_2$ distance | $\mathbf{73.34} \pm 3.55$ | $\mathbf{81.58} \pm 0.85$ |

Table 1: Test accuracy (%) when using weighted ("w.") similarity scores, against using normal similarity scores with different sentence transformers. The experiments are averaged over five runs and the standard error is reported. The best performing method within a weighted and non weighted similarity measure is marked in bold.

with different random seeds, and the mean and standard deviation thereof are reported. In the second experiment, we calculated the test set confidence intervals (CIs) through 1'000-fold bootstrapping as a proxy, due to the excessive computational cost.

## 4.1 Learned Weighted Similarity Measure

To explore the impact of the weighted embedding dimensions in the similarity measure, we carry out an experiment in which we compare the performance of the weighted with the unweighted model on the Rotten Tomato (RT) movie review dataset (Pang et al., 2002), as well as the AG news dataset (Zhang et al., 2016). The test accuracy results are displayed in Table 1. The results demonstrate that in 13 out of the 16 experiments, the model profits in performance from focusing on the task-relevant sentence embedding dimensions. The largest change can be observed when using standard cosine similarity on the AG news dataset with an accuracy of 78.86% ($\pm 0.21$ %p) increasing by almost 8%-points to 86.18% ($\pm 0.12$ %p) when using learned embedding dimensions. We observe, that for both datasets, the unweighted as well as the learned weighted cosine similarity performed better than $\ell_2$ or learned weighted $\ell_2$ similarity. Thus, for the remaining experiments, we will focus on the model using weighted cosine similarity.

| Test sample and its keywords | Prototype 1 and its most important words |
|---|---|
| ● This ==insufferable== ==movie== is meant to make you think about existential suffering. Instead, it'll ==only== put you to ==sleep==. | ● Plodding, ==poorly== written, murky and weakly ==acted==, the picture feels as if ==everyone== making it ==lost== their ==movie== mojo. |
| ● Because of an ==unnecessary== and ==clumsy== last scene,' swimfan' left me with ==a== very ==bad== feeling. | ● ==Plodding==, ==poorly== written, murky and ==weakly== acted, the picture feels as if everyone making it ==lost== their movie mojo. |
| ● Campanella's ==competent== direction and his ==excellent== cast ==overcome== the obstacles of a predictable outcome and a ==screenplay== that glosses over ==rafael's== evolution. | ● ...==bright==, ==intelligent==, ==and== humanly ==funny== film. |
| ● Between them, de niro and ==murphy== make showtime the ==most== savory and ==hilarious== guilty ==pleasure== of many a recent movie season. | ● ...==bright==, ==intelligent==, and humanly ==funny== ==film==. |

Table 2: Exemplary test samples from the movie review dataset by Pang et al. (2002) with the corresponding most important prototype from training. The most important words as determined by our model are visualized in ==color==. The highlighted words within the test sentence are extractive rationales, while the ones within the prototype can be thought of as abstractive rationales.

## 4.2 Visualizing Important Words

To demonstrate the effectiveness of our visualization approach, we first show qualitative samples from the test set of the movie review dataset (Pang et al., 2002). Table 2 presents test set sentences with prototypes and the corresponding most important words from the sentence itself (extractive rationales), but also from the prototype (abstractive rationales). Thanks to our innovation, we go beyond simply visualizing the prototypical sentence and show the most important words from the prototype as well as from the sample we are comparing it to. We enhance the interpretability mechanism, as we can show to which words from the prototype the test sample is most similar to and which words from it contributed most to the similarity. Hence, the most significant words from the same prototype can vary depending on the test samples, since the rationale for their similarity may differ.

## 4.3 Faithfulness

To evaluate the faithfulness of our model's visualization mechanism, we adopt the metrics of comprehensiveness and sufficiency, following the approaches of Jacovi and Goldberg (2021) and DeYoung et al. (2020). These metrics serve as proxies for assessing whether the visualized important words, referred to as rationales, faithfully account for the model's behavior. Sufficiency is quantified as the difference between the probability of the predicted class for the original input and the probability of the same class when making predictions solely based on the rationales. It measures the extent to which the extracted rationales are sufficient for a model to make its prediction (DeYoung et al., 2020). Comprehensiveness is similarly computed, but instead of considering the probability obtained with only the rationales, it subtracts the predicted probability of the input without any rationales. Comprehensiveness captures the degree to which the extracted rationales capture all tokens the model uses for prediction.

The previous work of Friedrich et al. (2022) lacks the extraction of specific rationales for test sentences. Thus, we employ the recurrent convolutional neural network (RCNN) of Lei et al. (2016) as a baseline that extracts rationales at the word-level from the test samples. Similarly, for our approach, we extract the important words by focusing on the similarity to the three prototypes that contribute most to the prediction from the test samples. The results of the evaluation on the Rotten Tomato movie review dataset (Pang et al., 2002) are presented in Table 3. Therefore, in this experiment we focus solely on extractive rationales to compute the comprehensiveness and sufficiency.

| Model | Comp. (%p)↑ | Suff. (%p)↓ | Accuracy (%) |
|---|---|---|---|
| Rationales RCNN (Lei et al., 2016) | 19.12 | 5.81 | 73.84 |
| 95%-CI | (18.20, 20.01) | (5.07, 6.58) | (71.97, 75.75) |
| Ours (BERT) | **52.90** | **2.63** | **82.65** |
| 95%-CI | (51.79, 54.12) | (1.63, 3.54) | (80.98, 84.33) |

Table 3: Comprehensiveness (Comp.), Sufficiency (Suff.), and Accuracy results on the movie review dataset, including bootstrapped 95% confidence intervals of the test set in gray.

Both models exhibit notably low sufficiency, with our method achieving a value of 2.63% points (95%-CI: 1.63%p, 3.54%p) and the baseline model achieving 5.81% points (95%-CI: 5.07%p, 6.58%p). These findings suggest that our proposed network extracts rationales that are utilized for prediction more sufficiently than the baseline. Additionally, our model demonstrates a comprehensiveness score that is more than 33% points higher than the baseline model. This observation suggests that while both methods extract relevant rationales, only our method visualizes the full set of rationales on which the model bases its prediction.

## 5   Conclusion

Our work presents several significant contributions. Firstly, we advance the research on extending prototypical networks to natural language processing by introducing a learned weighted similarity measure that enables our model to focus on the crucial dimensions within sentence embeddings while disregarding less informative context. This allows us to reap the benefits of pre-trained language models while retaining an inherently interpretable model. This enhancement leads to accuracy improvements over the previous work by Friedrich et al. (2022) on both the RT movie review dataset and the AG news dataset. Additionally, we propose a post-hoc visualization method for identifying important words in prototypical sentences and in test sentences. Our results demonstrate improved faithfulness in capturing the essential factors driving the model's predictions compared to Lei et al. (2016). Collectively, our findings highlight the efficacy of learned weighted similarity measure, the value of our post-hoc visualization technique, both leading to the enhanced faithfulness of the proposed model. These advancements contribute to a deeper understanding of the interpretability of neural networks in natural language processing tasks.

## Acknowledgements

MV is supported by the Swiss State Secretariat for Education, Research and Innovation (SERI) under contract number MB22.00047.

## Limitations

This study has certain limitations. Firstly, our focus is on two well-balanced classification datasets, and our performance is not competitive with non-interpretable state-of-the-art classifiers. However, our primary objective in this work is not to emphasize predictive performance compared to larger and more complex models, but rather to explore the interpretability mechanism of prototypical networks in the context of NLP.

Secondly, although prototypical networks exhibit an inherently interpretable network architecture, they still rely on a black box encoder. As demonstrated by Hoffmann et al. (2021), manipulating the model inputs can lead to interpretations that may not align with human understanding. Nonetheless, our research represents a significant step towards achieving explainable and interpretable AI.

Finally, the proposed post-hoc method for quantifying word-level importance is computationally expensive. It involves iteratively removing individual words from an embedding, which necessitates multiple forward passes through the network.

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

# A    Appendix

## A.1    Supplemental Related Work

Various methods have been proposed to address the explainability and interpretability of deep neural networks. In this section we are discussing the existing work, focusing especially on post-hoc explainability, rational-based interpretability, and prototype-based interpretability.

### A.1.1    Post-hoc Explainability

Post-hoc explanation methods are focused on finding correlations based on the inputs and outputs. Thus, explanations are commonly only created after forward passes of black-box models. One such method is e.g. to visualize the gradients of neural networks over the inputs with heat maps (Sundararajan et al., 2017; Smilkov et al., 2017). Alternatively, what has recently become popular with the rise of transformer architectures in NLP, are post-hoc explanations based on attention mechanisms (Bahdanau et al., 2015). For example Li et al. (2017) use the attention layers to detect important words in sentiment analysis tasks. However, Jain and Wallace (2019) point out some shortcomings in the explainability of attention heads and thus need to be handled with caution.

### A.1.2    Rational-based Models

Lei et al. (2016) introduced so-called rationals. These are pieces of the input text serving as justifications for the predictions. They used a modular approach in which a selector first extracts parts of a sentence followed by the predictor module which subsequently makes classifications based on the extracted rationals. Both modules are trained jointly in an end-to-end manner. They argue that using the selector, this fraction of the original input should be sufficient to represent the content. Nonetheless, this selector-predictor modular approach has several drawbacks (Jacovi and Goldberg, 2021). Jacovi and Goldberg (2021) argue that by training in this end-to-end fashion the selector may not encode faithful justifications, but rather "Trojan explanations". These may be problems like the rational directly encoding the label. Another illustrated failure case is that the selector module makes an implicit decision to manipulate the predictor towards a decision, meaning that the selector dictates the decision detached from its actual purpose to select important information for the predictor by extracting a rationale that has a strong class indication

it deems to be correct. For these reasons, Jacovi and Goldberg (2021) suggest a predict-select-verify model in which the model classifies based on the full input text but the selected rationale should in itself be sufficient to get to the same prediction, building up the extracted rational until this is the case.

### A.1.3 Prototypical Models

Prototype-based networks have recently been introduced for computer vision tasks to improve interpretability. In the paper "*This* Looks Like *That*: Deep Learning for Interpretable Image Recognition" (Chen et al., 2019), the authors argue that their ProtoPNet has a transparent, human-like reasoning process. Specifically, a ProtoPNet tries to find similarities between parts of an input image and parts of training images, called prototypes (hence the title of their paper), and classifies the input image based on this. Figure 1 shows an example for the classification of birds.

ProtoPNets have already been discussed in the setting of NLP by Friedrich et al. (2022) and Ming et al. (2019). Friedrich et al. (2022) change the encoder of the network from a convolutional neural network to a pretrained attention-based language model. Furthermore, they enhance the model with additional losses in the prototype layer to receive better prototypes on a word and sentence-level prediction. Ultimately, their so-called *i/Proto-Trex* is applied to classification and explanatory interactive learning (XIL). Ming et al. (2019) apply the concept of prototypes to sequential data and focus on different variants of RNNs. They identify that for sequential data it is often the case that multiple prototypes are similar. A solution they introduce an additional diversity loss that penalizes prototypes that are close to each other. However, RNNs have the problem that it is difficult to project the prototypes only to the relevant parts of a sequence. To overcome this limitation they use CNNs that can easily capture subsequences of an input. Further details will be explained in Section 3. Like with post-hoc and rational-based methods, prototypes have their shortcomings in their interpretability mechanism (Hoffmann et al., 2021). If there are subtle differences in the way the test and training data is acquired, it can lead to misguiding interpretations (such as JPEG compression for images).

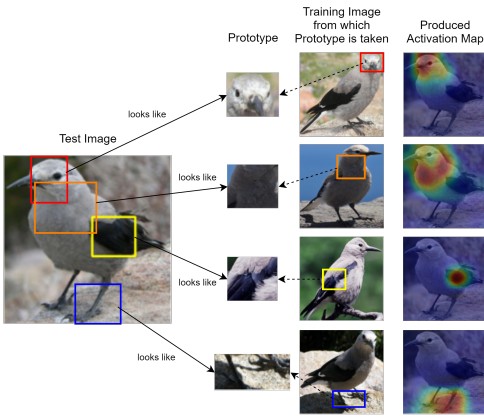

Figure 1: An example depicting the reasoning produced by ProtoPNet for computer vision.

### A.2 Experimental Setup

In the first experiment, where we examine the learned weighted similarity measure, we employed the 'bert-large-nli-mean-tokens' variant of Sentence-BERT, following the code provided by Friedrich et al. (2022). For the MPNET, we utilized the 'all-mpnet-base-v2' sentence transformer; for RoBERTa, we employed the 'all-distilroberta-v1' variant; and for GPT2, we utilized the pretrained transformer model 'Muennighoff/SGPT-125M-weightedmean-nli-bitfit' (all of which are available in the Hugging Face library). Across all training runs, we adopted the ADAM optimizer with a learning rate of 0.005 and a weight decay of 0.0005. Our models underwent 100 epochs of training, with a batch size of 128. We implemented a learning rate scheduler that reduced the learning rate by a factor of 0.5 every 30 epochs if there was no improvement in the validation loss. Similar to Friedrich et al. (2022), our model lacks an early stopping mechanism, as the projections only commence after 50% of the epochs have passed. A projection step, involving the projection of a prototype back to a training sample, was conducted every 5th epoch. In the final three epochs, only the fully connected layer was trained using positive weights to prevent negative reasoning, aligning with Friedrich et al. (2022)'s code implementation. Recognizing the impact of vector magnitudes on $\ell_2$ distance compared to cosine similarity, we adjusted the dimension weighting vector by applying a sigmoid function and then multiplying it by two, constraining the values to the range of (0, 2). For each class, we defined 10 prototypes, leading to a total of 20 prototypes for the movie classification and 40 for the AG News dataset. Furthermore, the

encoder layers are frozen during training, therefore the gradient only flows through the fully connected layer and the prototypical layer.

## A.3 Interpretability vs. Performance

As there is often a trade-off between the performance and the interpretability of deep neural networks, we analysed the predictive performance of a standard BERT on the movies and AG News dataset. Instead of appending a prototype layer after the transformer backbone, we directly added a fully connected (FC) layer to the classification output. We trained it with the cross entropy loss, and kept the whole experimental setup the same as in the previous experiments. The results are show in Table 4. We see that while the non-interpretable baseline performs better as expected, as it does not have underlying architectural constraints, the difference is not as prominent. Additionally, one could trade-off interpretability and accuracy by allowing for more prototypes, which would increase accuracy but make the method less interpretable as the simutability decreases (Lipton, 2018).

| | | Dataset | |
|---|---|---|---|
| **Backbone** | **Final Layer** | **RT reviews** | **AG News** |
| BERT | FC | **85.12** ± 0.25 | **90.29** ± 0.15 |
| | Prototype | 82.14 ± 0.20 | 86.38 ± 0.09 |
| GPT2 | FC | **77.87** ± 0.22 | **89.47** ± 0.14 |
| | Prototype | 75.44 ± 0.05 | 87.25 ± 0.13 |
| MPNet | FC | **85.12** ± 0.25 | **89.35** ± 0.05 |
| | Prototype | 79.55 ± 0.82 | 88.09 ± 0.28 |
| RoBERTa | FC | **84.37** ± 0.08 | **89.14** ± 0.05 |
| | Prototype | 77.51 ± 0.29 | 87.45 ± 0.07 |

Table 4: Test set accuracy (%) comparison of the performance vs. interpretability trade-off, by removing the prototypical layer and connecting the embedding layer directly to the classification output with a fully connected (FC) layer. The results are averaged over five runs, and the standard error is reported in gray. The best performing variant of a model is denoted in bolt.

## A.4 Loss Ablation Study

Here, we analyze the importance of the losses that we and Friedrich et al. (2022) are using:

$$\mathcal{L} = \frac{1}{n} \sum_{i=1}^{n} \text{CE}(t_i, y_i) + \lambda_1 \text{Clst}(\mathbf{z}, \mathbf{p})$$
$$+ \lambda_2 \text{Sep}(\mathbf{z}, \mathbf{p}) + \lambda_3 \text{Dist}(\mathbf{z}, \mathbf{p})$$
$$+ \lambda_4 \text{Divers}(\mathbf{p}) + \lambda_5 ||\mathbf{w}||_1 \tag{3}$$

where $t_i$ is the prediction, $y_i$ is the true label, $\mathbf{z} \in \mathbb{R}^d$ are the latent variables of the sentence-embedding, $\mathbf{p} \in \mathbb{R}^d$ are the prototypes, and $\mathbf{w} \in \mathbb{R}^{C \times d}$ are the weights of the last layer. Furthermore, for the loss to be complete, following must uphold: $\lambda_i > 0, i \in \{1, ..., 5\}$. (Please note, that by the time of writing, the formulas of the losses in the paper of Friedrich et al. (2022), as well as some interpretations included some mistakes. However, in their code they are implemented correctly.) To evaluate whether each of the loss terms is necessary we carry out an ablation study. In Table 5, the results of our loss ablation study are presented. We show the accuracy of our best-performing model on the movie review dataset with the different loss terms individually removed ($\lambda_j = 0, j \neq i$) from the full loss term as well as the top $k \in \{0, ..., 3\}$ prototypes removed. This allows inspecting whether each loss term had its desired effect.

In general, we observe that for the baseline model with all loss terms included, the model largely relies on the top 2 prototypes, as without them accuracy drops to 27.89%.

The clustering loss, $\text{Clst}(\mathbf{z}, \mathbf{p}) : \mathbb{R}^d \times \mathbb{R}^d \longrightarrow \mathbb{R}$, should ensure that each training sample is close to at least one prototype whereas the separation loss, $\text{Sep}(\mathbf{z}, \mathbf{p}) : \mathbb{R}^d \times \mathbb{R}^d \longrightarrow \mathbb{R}$, should encourage training samples to stay away from prototypes not of their own class (Li et al., 2018). In the results, we can see that seemingly the full loss ($\lambda_i \neq 0, i \in \{1, ..., 5\}$) relies more on the top 2 prototypes, as accuracy is much lower for the model with the clustering term removed than for the model with the full loss term. This suggests that without this term the model may rely on many prototypes to get to an estimate instead of only on the most similar prototype as it should encourage.

Results after removing the separation loss term show a slightly higher accuracy of 83.37% and a stronger drop-off when removing the prototypes. The steeper drop-off may imply that it has the desired effect of separating prototypes of different classes appropriately.

The goal of the distribution loss, $\text{Dist}(\mathbf{z}, \mathbf{p}) : \mathbb{R}^d \times \mathbb{R}^d \longrightarrow \mathbb{R}$, is to ensure that each prototype is close to a sample in the training set. The removal of this loss does not seem to hurt the performance. Hence, we believe it is not necessary and could be removed. An explanation of this behavior might be that while theoretically the maximum is specified over the whole training set, in the code of Friedrich

|                                    | Top k prototypes removed |         |         |         |
| ---------------------------------- | ------- | ------- | ------- | ------- |
| **Removed loss term**              | **k = 0** | **k = 1** | **k = 2** | **k = 3** |
| Baseline (full loss)               | 82.46   | 80.16   | 62.15   | 27.89   |
| Clustering loss ($\lambda_1 = 0$)  | 82.56   | 58.03   | 38.91   | 29.61   |
| Separation loss ($\lambda_2 = 0$)  | 83.37   | 66.94   | 40.11   | 23.10   |
| Distribution loss ($\lambda_3 = 0$) | 82.99  | 82.70   | 78.77   | 57.21   |
| Diversity loss ($\lambda_4 = 0$)   | 82.41   | 80.21   | 56.49   | 25.11   |
| Weights' $\ell_1$ loss ($\lambda_5 = 0$) | 82.51 | 79.59 | 55.20 | 25.16   |

Table 5: Loss ablation study showing accuracy when removing different number of top prototypes and using our best model with weighted cosine similarity on the RT polarity dataset.

et al. (2022) that we adopted, the maximum is only taken over a batch. Requiring each prototype to be close to a training sample from each batch might be too much to ask for. Further research might be going into a relaxation of this loss.

The idea of the diversity loss, $\text{Divers}(\mathbf{p})$ : $\mathbb{R}^d \longrightarrow \mathbb{R}$, is to push prototypes from the same class apart to learn different facets of the same class (Friedrich et al., 2022). Compared to the baseline, the results show that indeed accuracy is higher when removing the top $k$ prototypes and the diversity loss term seems to have the desired effect of prototypes learning different facets of the same class.

Finally, dropping the weights' $\ell_1$-regularization term does seem to slightly worsen the performance when dropping 1 to 3 prototypes.