# OpenReview forum: "This Reads Like That: Deep Learning for Interpretable Natural Language Processing"
_EMNLP/2023/Conference — EMNLP 2023 Main_

### Official Review · Reviewer_pffd · 2023-08-04

**Typos Grammar Style And Presentation Improvements:** L.188
**Soundness:** 4

**Excitement:**

4: Strong: This paper deepens the understanding of some phenomenon or lowers the barriers to an existing research direction.

**Paper Topic And Main Contributions:**

The paper presents an adaptation of prototype-based learning from the computer vision domain to NLP, introduces a learned weighted similarity measure, and proposes a post-hoc explainability method that allows highlighting the most important words in an inference as well as the corresponding words in the matching prototype. The similarity measure model performs a type of disentanglement, by separating and aligning embedding factors relevant to each prototype. The important word visualisation approach, despite being coarse and computationally expensive, can also be a valuable source of information from the base models. The results indicate that the proposed approach can successfully extract the information claimed in an interpretable manner.

**Reasons To Accept:**

The interpretability matter is highly relevant to the NLP community. The paper is a straightforward read and easy to follow. The results point to qualitatively high interpretability, but the price to pay for that is not quite clear (see reasons to reject).

**Reasons To Reject:**

While it is clear and rightly pointed by the authors that this work is not competitive with non-interpretable state-of-the-art classifiers, its would be important to have at least one such SOTA baseline for a better understanding of the price being paid for the interpretability achievements and how that could be mitigated. This of course is being said on the principle of making the paper argument complete, as one could obtain such answer by examination of relevant literature.

**Reproducibility:**

4: Could mostly reproduce the results, but there may be some variation because of sample variance or minor variations in their interpretation of the protocol or method.

**Reviewer Confidence:**

3: Pretty sure, but there's a chance I missed something. Although I have a good feel for this area in general, I did not carefully check the paper's details, e.g., the math, experimental design, or novelty.

---

> ### Author Rebuttal · Authors · 2023-08-29
>
> We thank the reviewer for the interesting points  and comments. Our response is denoted in normal font, whereas the questions/comments are in **bold** font.
>
> **While it is clear and rightly pointed by the authors that this work is not competitive with non-interpretable state-of-the-art classifiers, its would be important to have at least one such SOTA baseline for a better understanding of the price being paid for the interpretability achievements and how that could be mitigated. This of course is being said on the principle of making the paper argument complete, as one could obtain such an answer by examination of relevant literature.**
>
> This is an interesting point, and frequently, there exists a trade-off between the performance and interpretability of deep learning models. The primary focus of this paper is to enhance interpretability. We acknowledge that our predictive performance might not be directly comparable to that of the current SOTA models, which lack these explainable mechanisms. Nonetheless, upon evaluating the predictive performance, it becomes evident that the predictions remain competitive and non-random.
>
> To address these concerns, we have included an additional experiment. In this experiment, we aim to illustrate the interpretability cost incurred by incorporating an interpretable mechanism, such as the prototype layers. We replaced the prototypical layer with a fully connected layer connected to the classification output. The results are presented below. As we observe, despite not achieving SOTA status, the introduction of this supplementary interpretability component results in a decrease of 3-6%p in accuracy, which is a reasonable price to be paid for increased interpretability. Note that one can adjust this tradeoff by using more prototypes which reduces model interpretability but increases accuracy.
>
> |                 |                      | Dataset                        |                                |
> |-----------------|----------------------|--------------------------------|--------------------------------|
> | **Backbone** | **Final Layer** | **RT reviews**            | **AG News**        |
> | S-BERT          | FC                   | **85.12** $\pm$ 0.25        | **90.29** $\pm$ 0.15 |
> |                 | Prototype            | 82.14 $\pm$ 0.20                 | 86.38 $\pm$ 0.09          |
> | S-GPT2          | FC                   | **77.87** $\pm$ 0.22        | **89.47** $\pm$ 0.14 |
> |                 | Prototype            | 75.44 $\pm$ 0.05                 | 87.25 $\pm$ 0.13          |
> | MPNet           | FC                   | **85.12** $\pm$ 0.25        | **89.35** $\pm$ 0.05 |
> |                 | Prototype            | 79.55 $\pm$ 0.82                 | 88.09 $\pm$ 0.28          |
> | RoBERTa         | FC                   | **84.37** $\pm$ 0.08        | **89.14** $\pm$ 0.05 |
> |                 | Prototype            | 77.51 $\pm$ 0.29                 | 87.45 $\pm$ 0.07          |
>
> **Table 1**: Comparison of the performance vs interpretability trade-off, by removing the prototypical layer and connecting the embedding layer directly to the classification output with a fully connected layer.
>
>
> Again, we thank the reviewer for the comments, and hope that we could satisfy some of these.

---

### Official Review · Reviewer_L24G · 2023-08-07

**Soundness:** 4

**Excitement:**

4: Strong: This paper deepens the understanding of some phenomenon or lowers the barriers to an existing research direction.

**Paper Topic And Main Contributions:**

The paper investigates a method for interpretable text classification. The method uses prototypes for classification. The prototypes themselves are constructed from training samples, giving them a straightforward text interpretation. The method can identify the most important words from the prototypes, further improving interpretability. The paper performs evaluations on two classification datasets, showing that the model can extract interpretable word sequences for the model's classification decisions.

**Questions For The Authors:**

A) It would be very useful to see the full set of prototypes and rationales that the method generated. The supplemental materials appear to contain this information in a difficult-to-use format.

**Reasons To Accept:**

- The method is simple, reasonable, and (to the best of my knowledge) novel.

- The method is general and can be applied straightforwardly to other models and classification tasks.

- The experiments are well designed, and the results are convincing (for the datasets that the method is tested on).

- The paper is very well written.

**Reasons To Reject:**

I like the paper overall, and I think the contribution is probably sufficient for a short paper. The concerns below could be addressed in follow-up work.

- The method is only evaluated on a single encoder model, and on two classification datasets. Experiments on a larger range of models/datasets would be necessary for the claims to be fully convincing.

- The paper does not evaluate the magnitude of interpretability tax associated with the method.

- The baseline model which is chosen for comparisons is quite old, predating BERT by several years. It is not clear how to interpret the current method's improvement over this baseline.

**Reproducibility:**

4: Could mostly reproduce the results, but there may be some variation because of sample variance or minor variations in their interpretation of the protocol or method.

**Reviewer Confidence:**

3: Pretty sure, but there's a chance I missed something. Although I have a good feel for this area in general, I did not carefully check the paper's details, e.g., the math, experimental design, or novelty.

---

> ### Author Rebuttal · Authors · 2023-08-29
>
> We thank the reviewer for the interesting points and comments. Our response is denoted in normal font, whereas the questions/comments are in **bold** font.
>
> **A) It would be very useful to see the full set of prototypes and rationales that the method generated. The supplemental materials appear to contain this information in a difficult-to-use format.**
>
> We apologize for the inconvenience caused by the usability issues with the visualization of the prototypes and their corresponding key words. We have made the necessary improvements in the code, resulting in the production of a clearer CSV file. Additionally, as the reviewer expressed interest in the prototypes themselves, we have implemented code to include an extra CSV file containing these prototypes. For your convenience, we present below the set of 20 prototypes derived from the weighted cosine model applied to the movie dataset. Please note, that here we are not highlighting the important words, as we have no reference test sample to compare the prototypes to.
>
> Prototypes Weighted Cosine on Movie Dataset:
>
> *positive prototypes*
> - "the real star of this movie is the score, as in the songs translate well to film, and it's really well directed."
> - "a distinguished and thoughtful film, marked by acute writing and a host of splendid performances."
> - "an invaluable historical document thanks to the filmmaker's extraordinary access to massoud, whose charm, cultivation and devotion to his people are readily apparent."
> - "very well - written and very well - acted."
> - "a very charming and funny movie."
> - ". . . bright, intelligent, and humanly funny film."
> - "a very well - made, funny and entertaining picture."
> - "certainly the performances are worthwhile."
> - "it's a wise and powerful tale of race and culture forcefully told, with superb performances throughout."
> - "jones has delivered a solidly entertaining and moving family drama."
>
> *negative prototypes*
> - "the documentary is much too conventional--lots of boring talking heads, etc .--to do the subject matter justice."
> - "the leads we are given here are simply too bland to be interesting."
> - "unfortunately, the picture failed to capture me . i found it slow, drab, and bordering on melodramatic."
> - "i was perplexed to watch it unfold with an astonishing lack of passion or uniqueness."
> - "even as lame horror flicks go, this is lame."
> - "there is no pleasure in watching a child suffer . just embarrassment and a vague sense of shame."
> - "a bland animated sequel that hardly seems worth the effort."
> - "this is a poster movie, a mediocre tribute to films like them!"
> - "too clunky and too busy ribbing itself to be truly entertaining."
> - ". . . a bland, pretentious mess."
>
>
> **Table 1**: Prototypes on the movie classification dataset, when using a weighted similarity measure. Please note, that here we are not highlighting the important words, as we have no reference test sample to compare the prototypes to.
>
> **The method is only evaluated on a single encoder model, and on two classification datasets. Experiments on a larger range of models/datasets would be necessary for the claims to be fully convincing.**
>
> Certainly, we agree with the reviewer's observation. In response, we have introduced three additional backbones to showcase our findings (namely RoBERTa, GPT2, and MPNET). Presented below are the results obtained using five distinct random seeds, along with the corresponding mean and standard error values. The experimental setup were kept the same throughout the experiments and will be included in the manuscript.
>
> |                 |                      | Dataset                   ||
> |-----------------|----------------------|---------------------------|--------------------------------|
> | **Backbone** | **Similarity**  | **RT reviews**       | **AG News**               |
> | S-BERT          | Cosine               | 80.46 $\pm$ 0.43          | 78.67 $\pm$ 0.21               |
> |                 | Weighted cosine            | **82.14** $\pm$ 0.20 | **86.13** $\pm$ 0.12      |
> |                 | $\ell_2$ distance    | 79.20 $\pm$ 0.36          | **73.36 $\pm$ 0.89      |
> |                 | Weighted $\ell_2$ distance | **80.02** $\pm$ 0.44 | 73.11 $\pm$ 0.82               |
> | S-GPT2          | Cosine               | 73.41 $\pm$ 0.41          | 85.77 $\pm$ 0.20               |
> |                 | Weighted cosine            | **75.44** $\pm$ 0.05 | **87.25** $\pm$ 0.13      |
> |                 | $\ell_2$ distance    | 69.39 $\pm$ 1.78          | 78.36 $\pm$ 1.46               |
> |                 | Weighted $\ell_2$ distance | **69.88** $\pm$ 0.46 | **79.36** $\pm$ 1.36      |
> | MPNet           | Cosine               | **81.09** $\pm$ 0.49 | 86.49 $\pm$ 0.32               |
> |                 | Weighted cosine            | 79.55 $\pm$ 0.82          | **88.09** $\pm$ 0.28      |
> |                 | $\ell_2$ distance    | 75.87 $\pm$ 4.21          | **79.84** $\pm$ 1.50      |
> |                 | Weighted $\ell_2$ distance | **77.69** $\pm$ 2.26 | 79.73 $\pm$ 1.38               |
> | RoBERTa         | Cosine               | 75.58 $\pm$ 2.37          | 86.24 $\pm$ 0.06               |
> |                 | Weighted cosine            | **77.51** $\pm$ 0.29 | **87.45** $\pm$ 0.07      |
> |                 | $\ell_2$ distance    | 72.57 $\pm$ 1.79          | 80.48 $\pm$ 0.90               |
> |                 | Weighted $\ell_2$ distance | **73.34** $\pm$ 3.55 | **81.58** $\pm$ 0.85      |
>
> **Table 2**: Performance comparison of using weighted similarity scores, against using normal similarity scores over different sentence transformers. The experiments are averaged over five runs and the standard error is reported. The best performing method within a weighted and non weighted similarity measure is marked in bold.
>
>
> **The paper does not evaluate the magnitude of interpretability tax associated with the method.**
>
> Given that this concern was raised by another reviewer as well, we have taken the initiative to incorporate an extra experiment in the appendix. In this experiment, we contrast the prototypical network structure with direct classification subsequent to the encoder (connecting a fully connected layer from the sentence embedding dimension to the classification outputs), thereby bypassing the interpretability mechanism. The results of this comparison can be found below for your reference:
>
> |                 |                      | Dataset                        |                                |
> |-----------------|----------------------|--------------------------------|--------------------------------|
> | **Backbone** | **Final Layer** | **RT reviews**            | **AG News**        |
> | S-BERT          | FC                   | **85.12** $\pm$ 0.25        | **90.29** $\pm$ 0.15 |
> |                 | Prototype            | 82.14 $\pm$ 0.20                 | 86.38 $\pm$ 0.09          |
> | S-GPT2          | FC                   | **77.87** $\pm$ 0.22        | **89.47** $\pm$ 0.14 |
> |                 | Prototype            | 75.44 $\pm$ 0.05                 | 87.25 $\pm$ 0.13          |
> | MPNet           | FC                   | **85.12** $\pm$ 0.25        | **89.35** $\pm$ 0.05 |
> |                 | Prototype            | 79.55 $\pm$ 0.82                 | 88.09 $\pm$ 0.28          |
> | RoBERTa         | FC                   | **84.37** $\pm$ 0.08        | **89.14** $\pm$ 0.05 |
> |                 | Prototype            | 77.51 $\pm$ 0.29                 | 87.45 $\pm$ 0.07          |
>
> **Table 3**: Comparison of the performance vs interpretability trade-off, by removing the prototypical layer and connecting the embedding layer directly to the classification output with a fully connected layer.
>
> We see that while the non-interpretable baseline expectedly performes better, as it does not have underlying architectural constraints, the difference is not as prominent. Additionally, one could tradeoff interpretability and accuracy by allowing for more prototypes, which would increase accuracy but make the method less interpretable as the simutability [2] decreases.
>
> [2] Zachary C. Lipton. 2018. The Mythos of Model Interpretability: In machine learning, the concept of interpretability is both important and slippery. Queue 16, 3 (May-June 2018), 31–57. https://doi.org/10.1145/3236386.3241340
>
>
> **The baseline model which is chosen for comparisons is quite old, predating BERT by several years. It is not clear how to interpret the current method's improvement over this baseline.**
>
> Unfortunately, due to the constraints of the rebuttal period, we could not implement more recent extractive rationale baselines, which we leave for future work. We would like to add that our work focusses on an inherently interpretable method, which is a different direction than post-hoc explainable transformer methods, thus, we do not compare to such works.

---

### Official Review · Reviewer_AbxD · 2023-08-09

**Soundness:** 3

**Excitement:**

3: Ambivalent: It has merits (e.g., it reports state-of-the-art results, the idea is nice), but there are key weaknesses (e.g., it describes incremental work), and it can significantly benefit from another round of revision. However, I won't object to accepting it if my co-reviewers champion it.

**Paper Topic And Main Contributions:**

This work proposes an extension for prototypical networks by introducing a weight matrix (that is learned during training) to capture similarities between the prototype and original embedding. The authors investigate the extension of the proposed weight matrix for two similarity metrics, namely, L2 distance and cosine similarity. In their experiments, they show that the additional weighting increases the performance on two datasets and finally, showcase the usefulness of their approach for visualizing important prototypes.

**Questions For The Authors:**

[A] How exactly does your experimental setup look like? What SBERT model did you use (there are lots of different models available for SBERT) as this affects your weight matrix? Did you perform any hyper-parameter tuning? What are your other training parameters (e.g., number of epochs, batch-size etc.)?

[B] Are your results averaged across several runs with different seeds?

[C] "Lack of rationales for test sentences" (line 279): but Friedrich et al (2022) do also evaluate word-level models instead don't they? Given that GPT-2 outperforms SBERT, what would then the advantage of your approach be?

[D] What prototypes would be generated by the approach proposed by Friedrich et al (2022)?

[E] Is there really no better baseline that has been published since Lei et al (2016)? What about the work from Jacovi and Goldberg (2021) that you mention?

[F] Will the code be published?

**Reasons To Accept:**

This work investigates an interesting topic and propose a useful extension that inherently induces interpretability on a token-level.

I assume the code will be published given that it is included in the supplementary material.

**Reasons To Reject:**

Much of the experimental setup is unclear. For instance, the used model is not reported, there is no reporting on the used hyper-parameters and the baseline seems rather weak. The submitted code indicates the authors used some bert (1024-dim) and roberta (768-dim) models but it is unclear which ones exactly (cf. the list of available models in [1]). One would have to guess the parameters from the .yaml files and hope for the same results.



[1] https://sbert.net/docs/pretrained_models.html

**Reproducibility:**

4: Could mostly reproduce the results, but there may be some variation because of sample variance or minor variations in their interpretation of the protocol or method.

**Reviewer Confidence:**

2: Willing to defend my evaluation, but it is fairly likely that I missed some details, didn't understand some central points, or can't be sure about the novelty of the work.

**Typos Grammar Style And Presentation Improvements:**

Style: I'd suggest to do some hedging throughout the paper. For instance, the statement 'extend prototypical networks to [NLP]' in the abstract in the context with mentioning computer vision gives an impression that you are the first to do so.

line 580: the -> that?

---

> ### Author Rebuttal · Authors · 2023-08-29
>
> We thank the reviewer for the insightful questions and critical comments. We address the questions separately in **bold** and our response is denoted in normal font.
>
> **Much of the experimental setup is unclear. For instance, the used model is not reported, there is no reporting on the used hyper-parameters and the baseline seems rather weak. The submitted code indicates the authors used some bert (1024-dim) and roberta (768-dim) models but it is unclear which ones exactly (cf. the list of available models in [1]). One would have to guess the parameters from the .yaml files and hope for the same results.**
>
> We apologize for any confusion that may have arisen due to the lack of clarity in our initial manuscript version. Please refer to the supplementary setup provided below, which will also be included in the manuscript's appendix:
>
> In the first experiment, where we examine the learned weighted similarity measure, we employed the `bert-large-nli-mean-tokens` variant of Sentence-BERT, following the code provided by Friedrich et al. (2022). For the MPNET, we utilized the `all-mpnet-base-v2` sentence transformer; for RoBERTa, we employed the `all-distilroberta-v1` variant; and for GPT2, we utilized the pretrained transformer model `Muennighoff/SGPT-125M-weightedmean-nli-bitfit` (all of which are available in the Hugging Face library). Across all training runs, we adopted the ADAM optimizer with a learning rate of 0.005 and a weight decay of 0.0005. Our models underwent 100 epochs of training, with a batch size of 128. We implemented a learning rate scheduler that reduced the learning rate by a factor of 0.5 every 30 epochs if there was no improvement in the validation loss. Similar to Friedrich et al. (2022), our model lacks an early stopping mechanism, as the projections only commence after 50% of the epochs have passed. A projection step, involving the projection of a prototype back to a training sample, was conducted every 5th epoch. In the final three epochs, the prototypes were projected and fixed, and only the fully connected layer was trained. We only allow for positive weights in the fully connected layer as well as the dimension weighting in order to prevent negative reasoning, aligning with Friedrich et al. (2022). Recognizing the impact of vector magnitudes on L2 distance compared to cosine similarity, we adjusted the dimension weighting vector by applying a sigmoid function and then multiplying it by two, constraining the values to the range of (0, 2). For each class, we incorporated 10 prototypes, leading to a total of 20 prototypes for the movie classification and 40 for the AG News dataset.
>
> **[A] How exactly does your experimental setup look like? What SBERT model did you use (there are lots of different models available for SBERT) as this affects your weight matrix? Did you perform any hyper-parameter tuning? What are your other training parameters (e.g., number of epochs, batch-size etc.)?**
>
> Thank you for pointing this out. I hope with the aforementioned response we could clarify our experimental setup. We shall add the text to our manuscript’s appendix
>
> **[B] Are your results averaged across several runs with different seeds?**
> No, within the initial manuscript, we did not compute averages across distinct runs or various seeds. Instead, we conducted test set bootstrapping to incorporate confidence intervals, effectively capturing the inherent aleatoric uncertainty within the data. However, in response to your feedback, we have subsequently re-executed our experiments on five separate occasions using diverse seeds, except for the extractive sufficiency and comprehensiveness, as it is very computationally intensive, thus we kept the bootstrapped test interval. Here you can see the results for the first experiment:
>
> |                 |                      | Dataset                   ||
> |-----------------|----------------------|---------------------------|--------------------------------|
> | **Backbone** | **Similarity**  | **RT reviews**       | **AG News**               |
> | S-BERT          | Cosine               | 80.46 $\pm$ 0.43          | 78.67 $\pm$ 0.21               |
> |                 | Weighted cosine            | **82.14** $\pm$ 0.20 | **86.13** $\pm$ 0.12      |
> |                 | $\ell_2$ distance    | 79.20 $\pm$ 0.36          | **73.36** $\pm$ 0.89      |
> |                 | Weighted $\ell_2$ distance | **80.02** $\pm$ 0.44 | 73.11 $\pm$ 0.82               |
> | S-GPT2          | Cosine               | 73.41 $\pm$ 0.41          | 85.77 $\pm$ 0.20               |
> |                 | Weighted cosine            | **75.44** $\pm$ 0.05 | **87.25** $\pm$ 0.13      |
> |                 | $\ell_2$ distance    | 69.39 $\pm$ 1.78          | 78.36 $\pm$ 1.46               |
> |                 | Weighted $\ell_2$ distance | **69.88** $\pm$ 0.46 | **79.36** $\pm$ 1.36      |
> | MPNet           | Cosine               | **81.09** $\pm$ 0.49 | 86.49 $\pm$ 0.32               |
> |                 | Weighted cosine            | 79.55 $\pm$ 0.82          | **88.09** $\pm$ 0.28      |
> |                 | $\ell_2$ distance    | 75.87 $\pm$ 4.21          | **79.84** $\pm$ 1.50      |
> |                 | Weighted $\ell_2$ distance | **77.69** $\pm$ 2.26 | 79.73 $\pm$ 1.38               |
> | RoBERTa         | Cosine               | 75.58 $\pm$ 2.37          | 86.24 $\pm$ 0.06               |
> |                 | Weighted cosine            | **77.51** $\pm$ 0.29 | **87.45** $\pm$ 0.07      |
> |                 | $\ell_2$ distance    | 72.57 $\pm$ 1.79          | 80.48 $\pm$ 0.90               |
> |                 | Weighted $\ell_2$ distance | **73.34** $\pm$ 3.55 | **81.58** $\pm$ 0.85      |
>
> **Table 1**: Performance comparison of using weighted similarity scores, against using normal similarity scores over different sentence transformers. The experiments are averaged over five runs and the standard error is reported. The best performing method within a weighted and non weighted similarity measure is marked in bold. Here, we added three additional backbones as an ablation study, to demonstrate our results also with different sentence transformers, requested by reviewer L24G. From the results in the table, we can see that weighting the dimensions in the similarity measure helps improving the performance in almost all cases.
>
>
> **[C] "Lack of rationales for test sentences" (line 279): but Friedrich et al. (2022) do also evaluate word-level models instead don't they? Given that GPT-2 outperforms SBERT, what would then the advantage of your approach be?**
>
> Thank you for pointing this out. Yes, they do indeed also evaluate word-level models. Nevertheless, the reason why we do not compare our model on sufficiency and comprehensiveness with Friedrich et al. (2022) is because of a slight difference of interpretation of “rationales” and how they calculated their metrics. More precisely, in our calculation we are using **extractive** rationales, whereas in Friedrich et al. (2022) are using a form of **abstractive** rationales (see Gurrapu et al, 2023, https://arxiv.org/pdf/2301.08912.pdf, Fig 2.)
> Friedrich et al. (2022) mention in their paper that “Since our prototypical explanations are already short sequences, removing the entire prototype is similar to removing the rationale in the explanation.” (5. Faithful Prototypical Explanations). How we understand this, is that to calculate sufficiency and comprehensiveness, they do not remove any parts from a test sentence, but rather remove a prototype that is originating from the training set. Hence, during a forward pass the network would still see the whole test sample, which is a form of abstractive rationales.
> In our proposed method, we can have both, abstractive and extractive rationales. Similar to Friedrich et al. (2022) we can just remove a prototype to calculate the sufficiency and comprehensiveness. However, with our proposed method, we can also have extractive rationales, which we focus on in the results. In order to have a comparable baseline, we followed De Young et al. (2020) baseline, when they proposed the ERASER baseline and the sufficiency and comprehensiveness metric.
>
> Additionally, note that Friedrich et al. (2022) require a priori assumptions on their word-level prototypes. That is, they have to fix the number of words per prototype, as well as their relative positioning (eg adjacent words / every second word). Our method on the other hand does not require such restrictive assumptions and instead learns the number of words as well as their relative positive by itself, thus, allowing for much more flexibel prototypes. We believe this is the main advantage of the proposed method over Friedrich et al. (2022).
>
>
> **[D] What prototypes would be generated by the approach proposed by Friedrich et al (2022)?**
>
> This is an interesting question, but maybe a bit difficult to display. We present the reviewer here below the prototypes of the weighted cosine similarity BERT model (ours) vs the prototypes of the cosine similarity BERT model (Friedrich et al. (2022)) on the movie review dataset. Please note, that here we are not highlighting the important words, as we have no reference test sample to compare the prototypes to. Nevertheless, compared to Friedrich et al. (2022), depending on the test sample we could highlight the most important words in the prototype, which the previous method was not able to.
>
> Prototypes Weighted Cosine Similarity (ours):
>
> *positive prototypes*:
> - "the real star of this movie is the score, as in the songs translate well to film, and it's really well directed."
> - "a distinguished and thoughtful film, marked by acute writing and a host of splendid performances."
> - "an invaluable historical document thanks to the filmmaker's extraordinary access to massoud, whose charm, cultivation and devotion to his people are readily apparent."
> - “a very charming and funny movie.”
> - “very well - written and very well - acted.”
> - ". . . bright, intelligent, and humanly funny film."
> - "a very well - made, funny and entertaining picture."
> - “certainly the performances are worthwhile.”
> - "it's a wise and powerful tale of race and culture forcefully told, with superb performances throughout."
> - “jones has delivered a solidly entertaining and moving family drama.”
>
> *negative prototypes*:
> - "the documentary is much too conventional--lots of boring talking heads, etc .--to do the subject matter justice."
> - “the leads we are given here are simply too bland to be interesting.”
> - "unfortunately, the picture failed to capture me . i found it slow, drab, and bordering on melodramatic."
> - “i was perplexed to watch it unfold with an astonishing lack of passion or uniqueness.”
> - "even as lame horror flicks go, this is lame."
> - “there is no pleasure in watching a child suffer . just embarrassment and a vague sense of shame.”
> - “a bland animated sequel that hardly seems worth the effort.”
> - "this is a poster movie, a mediocre tribute to films like them!"
> - “too clunky and too busy ribbing itself to be truly entertaining.”
> - ". . . a bland, pretentious mess."
>
>
> **Table 2**: Prototypes on the movie classification dataset, when using a weighted cosine similarity measure.
>
>
> Prototypes Cosine Similarity (Friedrich et al.):
>
> *positive prototypes*:
> - "the performances are uniformly good."
> - "the result is something quite fresh and delightful."
> - "brings an irresistible blend of warmth and humor and a consistent embracing humanity in the face of life's harshness."
> - ". . . very funny, very enjoyable . . ."
> - "it's a masterpiece."
> - "the film brilliantly shines on all the characters, as the direction is intelligently accomplished."
> - "a riveting documentary."
> - "everywhere the camera looks there is something worth seeing."
> - "a thoughtful, moving piece that faces difficult issues with honesty and beauty."
> - "moving and vibrant."
>
> *negative prototypes*:
> - "plodding, poorly written, murky and weakly acted, the picture feels as if everyone making it lost their movie mojo."
> - "the character of zigzag is not sufficiently developed to support a film constructed around him."
> - "doesn't add up to much."
> - "this extremely unfunny film clocks in at 80 minutes, but feels twice as long."
> - "anyone not into high - tech splatterfests is advised to take the warning literally, and log on to something more user - friendly."
> - "thoroughly awful."
> - "simplistic, silly and tedious."
> - "the comedy is nonexistent."
> - "horrible"
> - "banal and predictable."
>
>
> **Table 3**: Prototypes on the movie classification dataset, when using a cosine similarity measure (non-weighted).
>
>
> **[E] Is there really no better baseline that has been published since Lei et al (2016)? What about the work from Jacovi and Goldberg (2021) that you mention?**
>
> We opted to utilize the work of Lei et al. (2016) due to its specific focus on extractive rationales. As it has been proposed by De Young et al. (2020) as a reference for evaluating comprehensiveness and sufficiency on the ERASER benchmark, we deem it a reasonable baseline. Please note that we believe it would be unfair to compare to transformer architectures that only offer interpretability post-hoc, as our model architecture is inherently interpretable [1].
>
> [1] Rudin, C. Stop explaining black box machine learning models for high stakes decisions and use interpretable models instead. Nat Mach Intell 1, 206–215 (2019). https://doi.org/10.1038/s42256-019-0048-x
>
> **[F] Will the code be published?**
>
> Certainly! The code package, along with the data we employed, was already made accessible to the reviewers. Our code had previously been publicly available, but we opted to set the GitHub repository to private in adherence to the privacy period stipulated by EMNLP. Rest assured, we have intentions of reverting the repository to public status as soon as we are permitted to do so.
>
> Again, we thank the reviewer for the insights and comments, and hope that we could satisfy some of these.

---

### Meta-Review · Area_Chair_MK8R · 2023-09-18

**Recommendation:** 4

**Metareview:**

The authors extend the prototypical network approach to NLP to obtain an inherently interpretable model and propose a learnable weighting scheme for computing semantic similarity. They evaluate their approach on two text classification datasets (RT, AGNEWS) atop one encoder model (SBERT) and show that their proposed weighting scheme improves upon baseline similarity metrics. They further show that prototypical networks improve upon faithfulness — as estimated through comprehensiveness and sufficiency compared to a baseline.

The reviewers agree that the paper is well written and easy to follow (pffd, L24G). The reviewers further commend the simplicity, usefulness and relevance of the proposed method (AbxD, L24G, pffd).

The main concerns raised by the reviewers were (1) the lack of an analysis of the “interpretability tax” (L24G, pffd), (2) an unclear experimental setup (AbxD) and (3) a narrow experimental evaluation (L24G).
In the discussion period, the authors have addressed issues (1, 2, 3) with detailed comments and additional experiments, which they have committed to add to their paper — the author response has thus alleviated these concerns of the reviewers.

Another concern raised by the reviewers (L24G, AbxD) was the lack of relatively novel baselines, as the work of Lei et al is relatively outdated — as more novel baselines would help contextualize the contributions of the authors’ work. Some more recent and very relevant inherently interpretable methods which were not cited by the authors and can also be used as baselines are [1], [2]. While due to the speed of the field it is possible to miss relevant work, these papers are relatively well known in the field of explainable NLP and their omission is somewhat of a concern.

Furthermore, reviewer AbxD accurately notes that the paper might benefit from some hedging with respect to the strength and generality of claims made. Apart from reviewer AbxD’s comment, in the abstract (and in the conclusion) the authors claim that “… our proposed method not only improves predictive performance…” — however this improved predictive performance holds with respect to the baselines without the learned weighting scheme, and not in general. The experimental scope should be clearly delineated in scope of the paper.

The conclusion also does not mention the extension of prototypical networks to NLP, which should be mentioned as a contribution.

[1] Jain, Sarthak, et al. "Learning to faithfully rationalize by construction.” (2020)

[2] Bastings, Jasmijn, Wilker Aziz, and Ivan Titov. "Interpretable neural predictions with differentiable binary variables." (2019).

---

### Decision · Program_Chairs · 2023-10-07

**Decision:**

Accept-Main

**Comment:**

The authors extend the prototypical network approach to NLP to obtain an inherently interpretable model and propose a learnable weighting scheme for computing semantic similarity. They evaluate their approach on two text classification datasets (RT, AGNEWS) atop one encoder model (SBERT) and show that their proposed weighting scheme improves upon baseline similarity metrics. They further show that prototypical networks improve upon faithfulness — as estimated through comprehensiveness and sufficiency compared to a baseline.

The reviewers agree that the paper is well written and easy to follow (pffd, L24G). The reviewers further commend the simplicity, usefulness and relevance of the proposed method (AbxD, L24G, pffd).

The main concerns raised by the reviewers were (1) the lack of an analysis of the “interpretability tax” (L24G, pffd), (2) an unclear experimental setup (AbxD) and (3) a narrow experimental evaluation (L24G).
In the discussion period, the authors have addressed issues (1, 2, 3) with detailed comments and additional experiments, which they have committed to add to their paper — the author response has thus alleviated these concerns of the reviewers.

Another concern raised by the reviewers (L24G, AbxD) was the lack of relatively novel baselines, as the work of Lei et al is relatively outdated — as more novel baselines would help contextualize the contributions of the authors’ work. Some more recent and very relevant inherently interpretable methods which were not cited by the authors and can also be used as baselines are [1], [2]. While due to the speed of the field it is possible to miss relevant work, these papers are relatively well known in the field of explainable NLP and their omission is somewhat of a concern.

Furthermore, reviewer AbxD accurately notes that the paper might benefit from some hedging with respect to the strength and generality of claims made. Apart from reviewer AbxD’s comment, in the abstract (and in the conclusion) the authors claim that “… our proposed method not only improves predictive performance…” — however this improved predictive performance holds with respect to the baselines without the learned weighting scheme, and not in general. The experimental scope should be clearly delineated in scope of the paper.

The conclusion also does not mention the extension of prototypical networks to NLP, which should be mentioned as a contribution.

[1] Jain, Sarthak, et al. "Learning to faithfully rationalize by construction.” (2020)

[2] Bastings, Jasmijn, Wilker Aziz, and Ivan Titov. "Interpretable neural predictions with differentiable binary variables." (2019).